# Cultural Sustainability of US Cities: The Scaling of Non-Profit Arts Footprint with Population

**George P. Kraemer**

Department of Environmental Studies, Purchase College (SUNY), Purchase, NY 10577, USA;
george.kraemer@purchase.edu

**Abstract:** The functional characteristics of urban systems vary predictably with Metropolitan Statistical Area (MSA) population, with certain metrics increasing apace with population (e.g., housing stock), some increasing faster than population (e.g., wealth), and others increasing slower than population (infrastructure elements). Culture has been designated the fourth pillar of sustainability. The population-dependent scaling of operating revenue, work space, and number of employees was investigated for almost 3000 arts organizations in the US, both in aggregate and by arts discipline (music, theater, visual and design arts, dance, and museums). Unlike general measures of creativity, the three measures of economic footprint did not scale supra-linearly with the population of metropolitan areas. Rather, operating revenue scaled linearly (e.g., like amenities), and work space and employee number scaled sub-linearly (e.g., like infrastructure). The cost of living, proxied by housing costs, increased with MSA population, though not as rapidly as did arts organization operating revenue, indicating a degree of uncoupling. The generally higher educational attainment of adults in larger cities, coupled with the growth of the education-dependent arts patronage, suggest a funding focus on less populous (50,000–1,000,000), as well as on under-performing, cities.

**Keywords:** urban; arts; scaling; population; economic footprint

## 1. Introduction

Much of the global population growth over the next 30 years will occur in cities [1]. As the fraction of humans living in urban environments increases, so does the need for sustainable operation of these dense aggregations. Sustainable development and operation have traditionally included considerations of the ecological, economic, sociopolitical domains [2]. However, truly sustainable urban systems require more than the provision of resources and services, and the elimination of wastes. Culture has been identified as the fourth pillar of sustainability [3,4], an essential but under-studied pillar [5]. There is a general tendency to focus on the instrumental value of non-market services. However, beyond the environmental and socio-economic improvements, including culture in sustainable operations planning may have unexpected ancillary benefits; people in cities that rank higher on sustainability scales reported higher levels of happiness or satisfaction with life [6–10].

Per capita economic output is greater in urban systems in the United States (US), compared with output from outside; urban cores and the surrounding economically interacting areas are responsible for the majority of economic output on national scales, accounting for an estimated 73% of the gross domestic product of the US [11]. The relative economic dominance of cities is even greater in developing countries, where much of future growth is expected.

The growing trend toward urbanization makes cities increasingly important as drivers of environmental change on local to global spatial scales. For example, cities constitute only 3% of land area, but consume ca. 75% of delivered energy [5]. Hence, sustainability writ broadly will necessarily focus on urban environments and their surroundings. The

United Nation's SDG 11, Sustainable Cities and Communities, explicitly acknowledges the increasing value of cities and their locally interacting metropolitan areas.

Cities display unpredicted (i.e., emergent) properties, such as self-organization into neighborhoods. City organization and dynamics influence economy, on top of which are layered the influences of local history, geography and culture [12–14]. Population size is a major determinant of the dynamics of urban functional characteristics in cities across the globe [12,15]. Scaling analysis, which relates the characteristics and function of a system to its size, has demonstrated a robust connection between population and socio-economic activity. On average, cities with larger populations demonstrate economies of scale for infrastructure elements (i.e., road surface), are more productive than smaller cities, and show greater rates of wealth generation and intellectual creativity [15]. The fact that power law scaling can relate the characteristics of cities of a wide range (three orders of magnitude) of populations, both in the US and world-wide, suggests an underlying mechanism that is to an extent independent of the local idiosyncrasies of history, geography and culture [12,13].

Florida [16] argued that cities concentrate and generate creative occupations that encompass both the technological and cultural realms. However, cities do not all agglomerate talent equally; artist mobility fosters the existence of talent sinks, cities whose greater creativity and innovation increase their attractiveness to both artists and the general public [17]. The arts can serve as an economic engine through the provision of lodging and ancillary visitor services [3,18], with the general result of the enhanced economic vibrancy of cities. The socio-economic boost can be seen in the case of Philadelphia; those areas characterized by a strong arts presence experienced a more rapid reduction in poverty than surrounding neighborhoods [19]. The arts also help differentiate cities' personalities by strengthening the city's image and shaping its brand, which is perceived by consumers as important added value [20,21].

Bettencourt et al. [22] emphasized the need to identify urban organization and dynamics to aid the development of sustainable solutions for growing cities. Sustainability planning for future urban development will increasingly use big data (e.g., to improve resource use efficiency and general attractiveness [23]). Big data derived from urban systems have revealed patterns in everyday life, ranging in scale from short-term and small scale (e.g., cell phone traffic) to long-term and larger scale (e.g., re-development, population growth). Cultural and creative industries, including the arts, generate multiple benefits, and have therefore become an important focus of cultural, social, and economic policy. Economic stimulus packages investing in the arts can revitalize regional economies [24]. The proximity to cultural amenities can significantly elevate regional growth, nurtured by increases in "high human capital" (i.e., highly educated residents) [25]. This human capital feeds back positively on the future development of arts institutions, more strongly even than economic output [26].

As part of an effort to understand the role of arts organizations in future sustainable development and to help improve the efficiency of resource allocation, this study determined the scaling relationships for (i) operating revenue, (ii) work space, and (iii) number of employees with metropolitan statistical area (MSA) population. MSAs, the unit of analysis in the study, are useful for understanding large-scale function (e.g., [12,13]), and are relevant to considerations of the economic footprint of the arts. Other studies have used MSAs as the unit of analysis; Li [26] used them in connecting the cultural economy and city competitiveness.

This research bridges the gap between an easily measured variable (population) and predictions that may guide more efficient use of resources. Mid-size and large cities were compared as cultural repositories for four arts disciplines (music, visual and design arts, theater, dance) and museums, the sub-CCI units Li [26] identified as in need of more study. The density of arts opportunities for urban dwellers and the economic footprint of non-profit arts institutions were also compared for large (>1 M residents) and mid-sized (50 K–1 M residents) MSAs.

## 2. Materials and Methods

The analysis of the scaling of US city size and characteristics relied on a combination of several data sources. SMU DataArts [27] provided data that richly detail the operational characteristics of non-profit, grant-funded arts organizations within the US. This dataset captures most of the funding (R. Johnson, [28]) received by non-profit arts organizations. Since government support of US arts organizations tends to be less than in other countries, philanthropic sources are correspondingly more important [29].

The data analyzed in this paper were collected in 2019. While this database is updated annually, 2019 data were chosen as a pre-pandemic baseline, against which we could examine future funding, so as to define COVID-related impacts. The data were disaggregated based on self-reported NISP (National Standard for Arts Information Exchange Project) classification [30] into five disciplines (music, dance, theater, visual and design art, and museums), plus a full arts category that included the five subdisciplines plus photography, crafts, media arts, and literature. For 2019, 2968 arts organizations were represented in the full arts category, while the organizations in the focal disciplines ranged from 184 (museums) to 642 (music). The arts organizations represented a range of sizes, but were mostly moderate; 71% reported annual operating revenues of USD 10,000–1,000,000 (median = USD 392,000), and 77% having 10–100 full time employees (median = 21).

The US Census Bureau data provided 2019 populations of MSAs in the United States. MSAs are county-based regions containing at least one urbanized area surrounded by a socially and economically integrated area. The US Census Bureau also provided measures of educational attainment (number of adults over 25 years with a bachelor degree or greater) via the American Community Survey dataset. Cities listed in the SMU DataArts database were compared against the list Census Bureau MSA populations. Cities receiving arts funding but not listed as an MSA were examined individually; arts organizations in cities that were within a defined MSA (e.g., Berkeley, CA within the San Francisco MSA) were re-coded as part of the parent MSA.

The overall cost of urban living is strongly influenced by housing costs [31]. As a proxy for the cost of living, 2019 real estate values for mid-tier (35th–65th percentile) single family residences were obtained from Zillow.com. In addition, fair market rent estimates for 1-bedroom apartments in metro areas were summarized using data drawn from U.S. Department of Housing and Urban Development.

This analysis focuses on three metrics that help define the economic and cultural footprint of arts organizations: operating revenue, gross work space, and number of employees. All organizations reported operating revenue, with 82% and 96% return rates for work space and employee number, respectively. Organization locations were checked against the MSAs defined by the Census Bureau. Organizations from locations with populations less than 50,000, and those reporting zero operating revenue, zero employees, or work space less than 100 ft$^2$, were removed prior to analysis of each metric.

Prior studies have demonstrated that MSA functional characteristics can be represented as power law functions of population size [12,13]—$Y = Y_0 \times P^{\beta}$, where $Y$ is the modeled characteristic, $Y_0$ is a scaling constant, $P$ = MSA population, and $\beta$ is the scaling exponent. The $\beta$ value characterizes the sensitivity of a metric to increases in population size (i.e., the metric's elasticity). A $\beta$ value < 1 (sub-linear scaling) is typically associated with infrastructural economies of scale (e.g., road surface area). A $\beta$ value = 1 (linear) describes population needs and amenities (e.g., number of dwellings), while $\beta$ > 1 (supra-linear scaling) characterizes generally productive measures, such as gross domestic product or patent production, which increase faster than population. MSA population was plotted against the sum of each metric for that MSA, and a two-factor power model was used to estimate $\beta$, as well as the associated error of the estimate. Scaling exponents were then tested against $\beta$ = 1 (linear scaling) to determine the class of behavior (i.e., sub-linear, linear, or supra-linear).

Scale-Adjusted Metropolitan Indicators (SAMIs; [32]) were calculated for $\beta$ values for the full arts category, and for each of the subdisciplines. SAMI values are dimensionless

and independent of MSA population, land area, and population density. They allow comparisons of the characteristics or functional performance of individual MSAs to identify those that over- or under-perform relative to what is expected for an MSA of a given size. SAMIs were calculated for each MSA for each metric as:

$$\text{SAMI} = \ln\left(\frac{\text{observed metric value}}{\text{value expected of a city of its size}}\right)$$

where the expected value was estimated using the power relationship obtained above. A SAMI value of 0 indicates average performance as predicted by the full dataset for a particular MSA population. SAMIs were ranked, and the cities with the 10 most positive and most negative values (indicating over- and under-performance, respectively) were recorded. SAMIs may be interpreted as depicting the relative influence of local characteristics (history, geography, culture, etc.) on the performance metric, an influence outside the effect of population size [13].

Scatterplots of population vs. MSA metric (e.g., operating revenue, educational attainment) were analyzed via a two-parameter power regression of metric on population. Arts organization density (no. $10^{-6}$ residents) and population sizes of MSAs with the largest and smallest SAMI values were not normally distributed; organization densities and MSA populations were compared using the Mann–Whitney U test.

## 3. Results

The operating revenues of all arts organizations in the dataset were scaled linearly with population size (Figure 1, top panel), as β was not significantly different from 1. Removing outliers (i.e., values greater than two standard deviations from the value expected based on population size) did not significantly change the scaling exponent. Working space and the number of employees both scaled sub-linearly with population (Figure 1, middle, bottom panels). Similarly, the removal of outliers did not change the scaling exponents. A 10-fold increase in city population was accompanied by a ca. 10-fold increase in total operating revenue, a 6-fold increase in total metropolitan work space, and a 5-fold increase in the total number of employees. The 10 MSAs with the largest positive SAMIs (i.e., the over-performing urban areas) were significantly larger (median population = 3,462,624) than under-performing cities (median = 223,960; Mann–Whitney U = 16, $p$ = 0.010).

SAMIs (Scale-Independent Metropolitan Indices) differed between specific arts disciplines (Figure 2). Music organizations were, by far, characterized by the largest spread between the top 10 over-performing metro areas (i.e., greater operating revenue that predicted based on population) and the bottom 10 under-performing areas. The same large range was not observed for the other arts disciplines or museums. Boston, Cleveland, and New York City were in the top 10 SAMIs for four of the five arts disciplines. Los Angeles and Pittsburgh were in the top 10 of three of the arts disciplines. Atlanta and Denver were both in the bottom 10 of three of the arts disciplines. Little Rock (AR), Oxnard (CA), and Oakland (CA) were in the bottom 10 of two of the arts disciplines.

The operating revenues of the four arts disciplines (music, theater, dance, visual and design arts) and museums scaled linearly with population (i.e., β value not significantly different from 1; Figure 3). With the exception of the working space of theater and visual and design arts, the total number of employees and work space of museums and arts disciplines scaled sub-linearly with population.

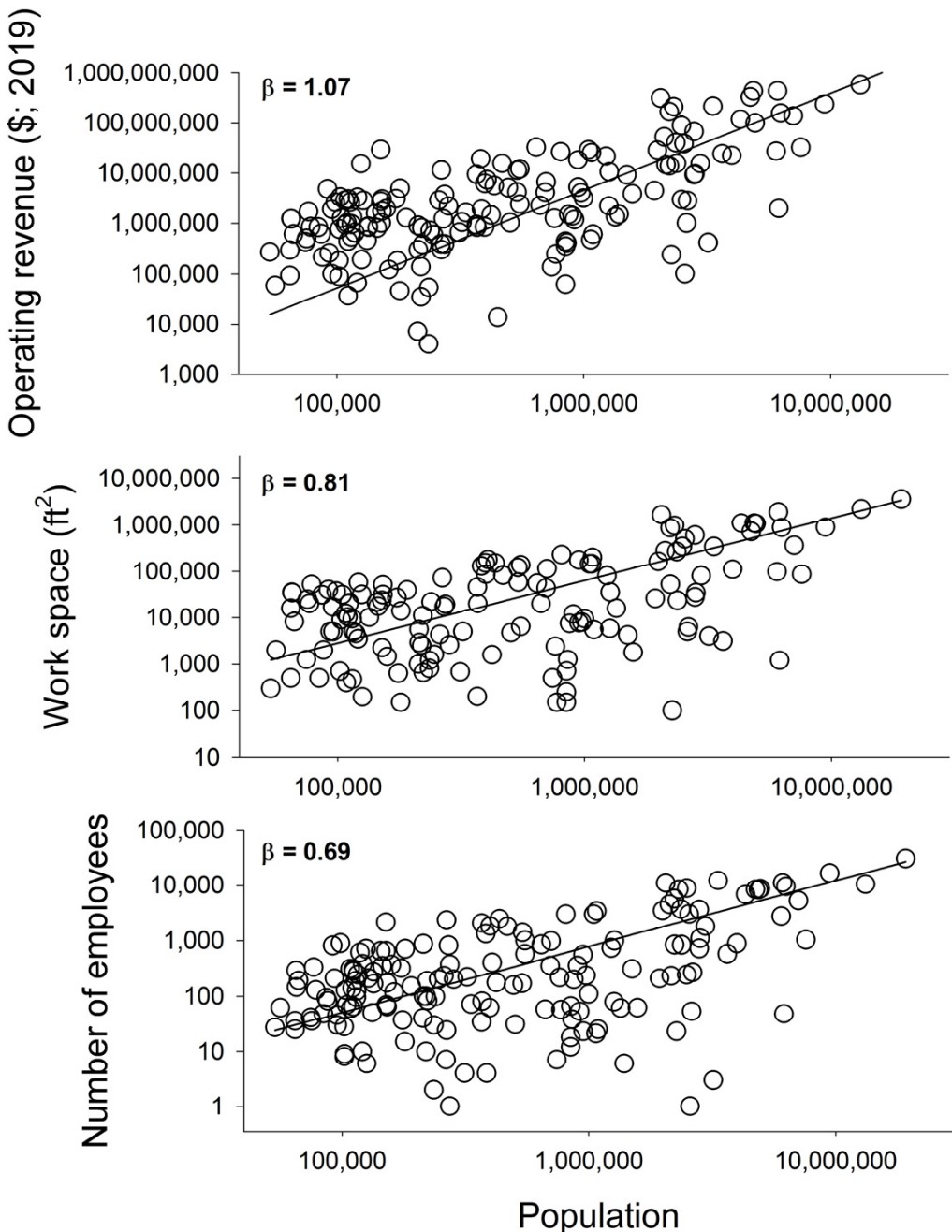

**Figure 1.** Scaling of cultural footprint metrics (operating revenue (**top panel**), work space (**middle panel**), and number of employees (**bottom panel**)) with population of metropolitan statistical area. All regression slopes are significantly different than 0 ($p < 0.001$). Operating revenue power exponent ($\beta = 1.07$) is not significantly different from 1. Exponents for space ($\beta = 0.81$) and number of employees ($\beta = 0.69$) are significantly less than 1.

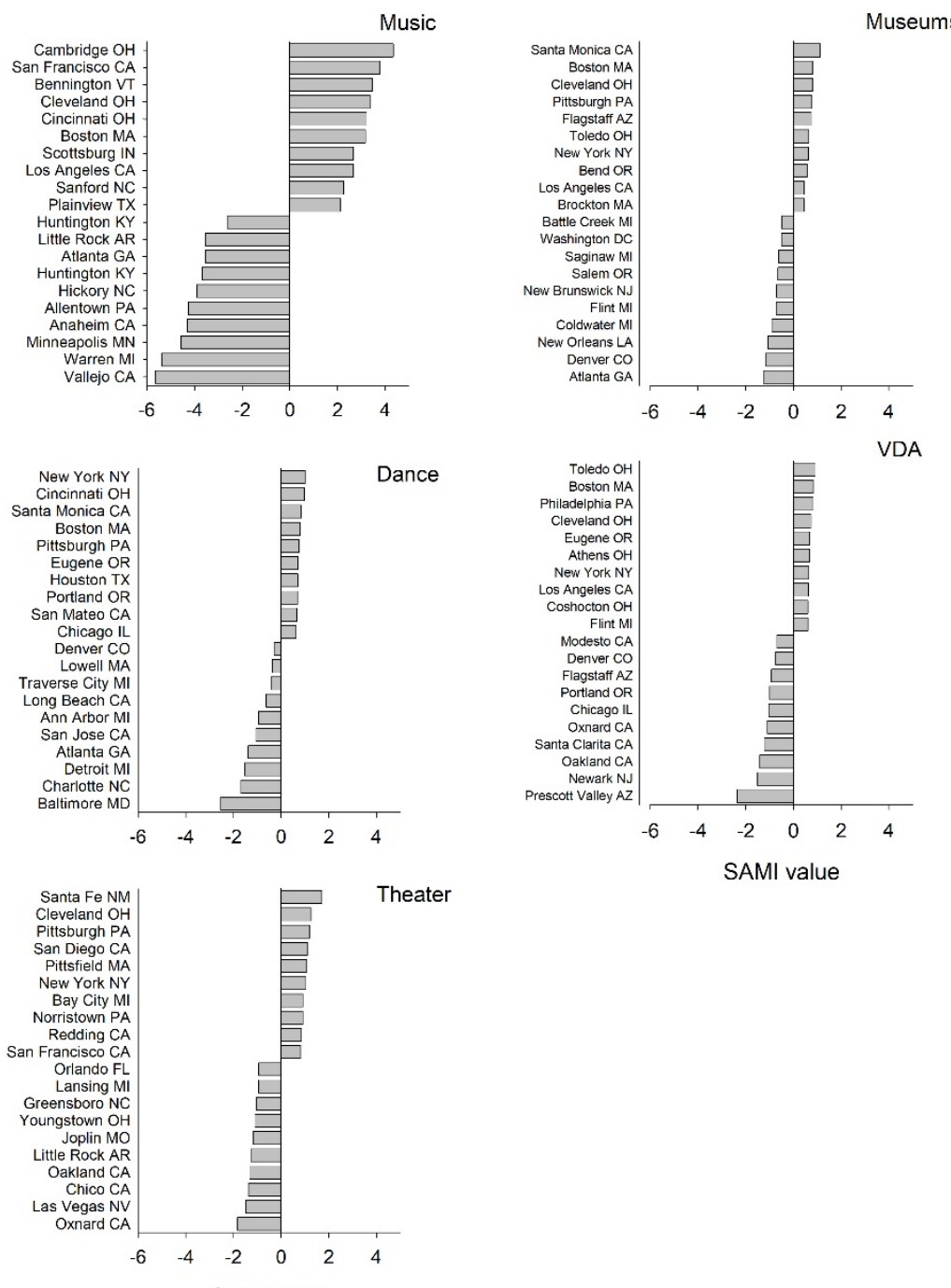

**Figure 2.** SAMI values (Scale Independent Metropolitan Index) for operating revenue of each arts discipline (VDA = visual and design arts). The magnitude of the SAMI value presents the difference between the reported revenue and the revenue expected for a metropolitan statistical area of that population.

Cost of living, defined by housing costs, was only mildly influenced by metro population. Real estate values for mid-tier single-family homes increased sub-linearly ($\beta = 0.11$) as MSA population increased ($p < 0.001$; Figure 4). Similarly, fair market rent estimates increased sub-linearly ($\beta = 0.13$) with population size ($p < 0.001$; Figure 5). A ten-fold increase in population predicted only a ca. 40% increase in housing costs, compared with a predicted 12-fold increase in operating revenue.

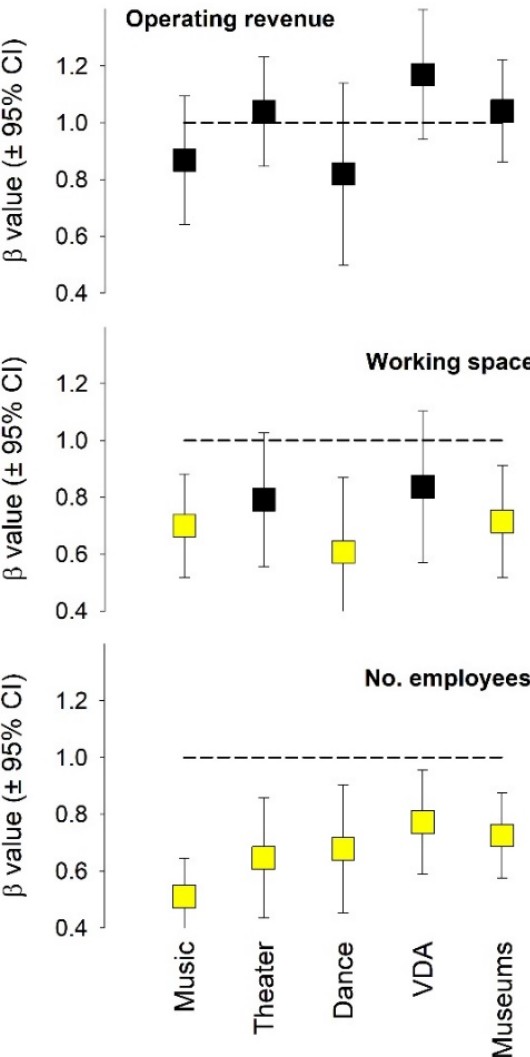

**Figure 3.** Values of β (scaling exponents) of cultural footprint metrics (operating revenue, work space, number of employees). Yellow symbols, with error bars below the dotted line, represent β values that are significantly less than 1 (i.e., scale sub-linearly).

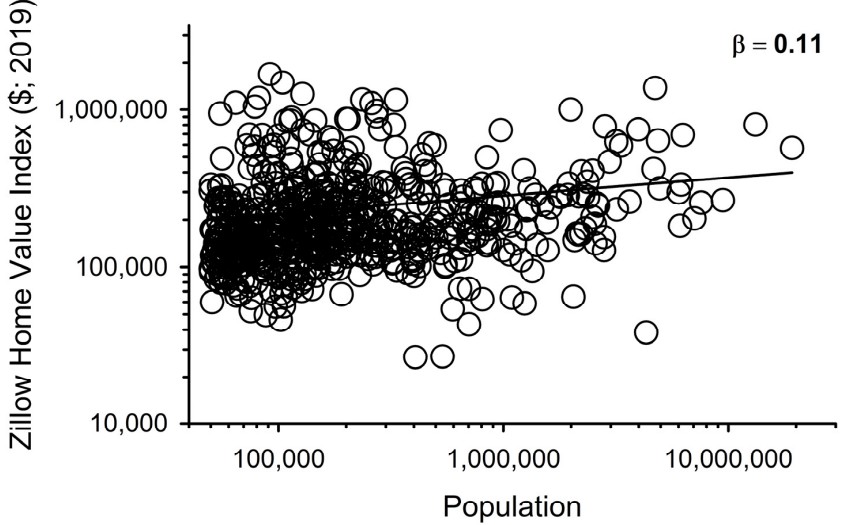

**Figure 4.** Regression of one mid-tier ZHVI (Zillow Home Value Index; 2019) on MSA (metropolitan statistical area) population ($p < 0.001$).

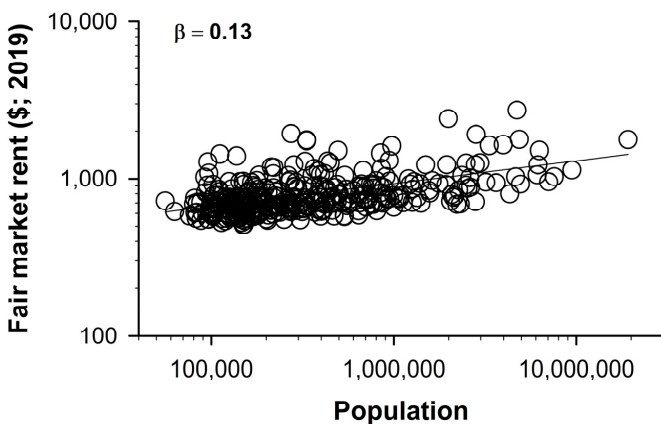

**Figure 5.** Regression of one-bedroom rental cost (2019) on metropolitan statistical area population ($p < 0.001$).

Mid-sized metropolitan statistical areas (50 K–1 M) were home to 60% (57–64%) of the non-profit arts organizations for music, visual and design arts, theaters, and museums. The notable exception was dance, for which only 17% of organizations were located in mid-sized MSAs. The density of arts organizations (number per million residents) varied as a function of discipline (Figure 6, top panel). Mid-size and large metro areas differed significantly in density of organizations, with the median densities of the two MSA population classes being roughly similar for music and theater. Museum density was 7-fold greater in mid-size metro areas. The organization density in mid-sized cities (50 k–1 M) was significantly greater for all arts disciplines than the density in large cities (>1 M; Mann–Whitney *p*-values < 0.001; Figure 6).

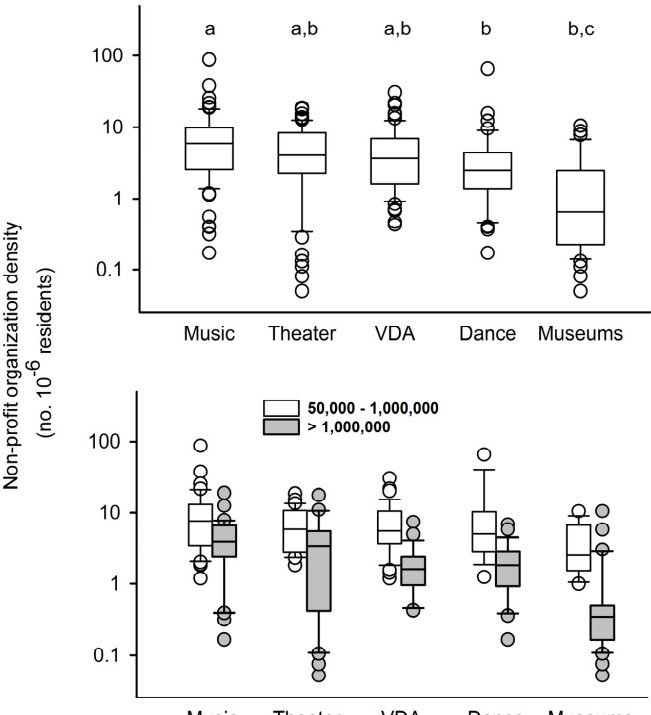

**Figure 6.** Density of non-profit arts organizations (no. $10^{-6}$ residents) as a function of arts discipline (**top panel**) and discipline x metro statistical area (MSA) population (**bottom panel**). Different letters signify statistically different densities (**top panel**). The densities of all disciplines were significantly higher in mid-sized MSAs than in large MSAs ($p < 0.01$). Different letters in the top panel signify significantly different values.

The total economic footprint, estimated as the sum of operating revenues, differed by arts discipline and metro area size (Table 1). Arts organizations in large metro areas were characterized by a total operating revenue ca. 17-times greater than the totals for mid-sized metro areas. Overall, museums, theater, and visual and design arts organizations were responsible for 80% of the total operating revenues of all arts disciplines in all metro areas. Music and dance accounted for only 15% and 5%, respectively, of the total operating revenues of non-profit arts institutions.

**Table 1.** Sum of operating revenues (USD; rounded to nearest 1000) of arts organizations as a function of metropolitan area population (mid-sized population: 50 K–1 M; large: >1 M).

| Discipline | Mid-Sized (USD) | Large (USD) |
| --- | --- | --- |
| Museums | 112,802,000 | 2,513,212,000 |
| Theater | 150,897,000 | 2,019,957,000 |
| Visual and design arts | 96,391,000 | 1,942,973,000 |
| Music | 99,542,000 | 1,231,900,000 |
| Dance | 12,906,000 | 441,288,000 |

Educational attainment, measured as the number of adults above 25 years old, scaled positively and supra-linearly ($\beta = 1.14$) with city population (Figure 7). These results were also apparent for the highest educational attainment; adults having earned a graduate or professional degree also scaled supra-linearly with MSA population ($\beta = 1.13$; not shown).

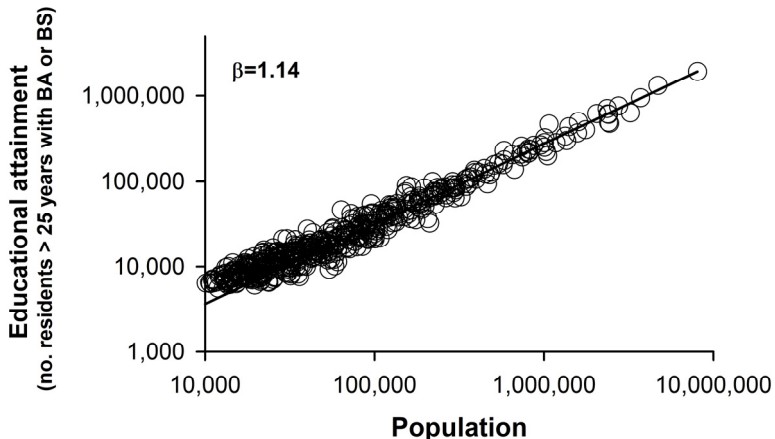

**Figure 7.** Educational attainment (number of residents >25 years old with bachelor degree) as a function of the populations of metropolitan statistical areas ($p < 0.001$).

## 4. Discussion

Sustainable operations, especially in increasingly populous urban systems, must include not only the traditional three pillars (balanced and equitable ecological function, economic viability, sociopolitical development), but also culture, the fourth pillar of sustainability (e.g., [2]). Culture helps define a city's identity, and arts represent a key cultural differentiator. For example, the Broadway theater district has long been a cultural identifier associated with New York City. Those cities rich in culture may also communicate a sense of safety and security [26,29], and engender a greater self-reported level of happiness [6–10]. Curtis et al. [33] argued that the arts support sustainability efforts via increased pro-environmental beliefs, values, and attitudes, by helping form pro-environmental social norms, and increasing community involvement in pro-environmental activities.

An unexpected outcome of the analysis was that the three metrics of the economic footprint of not-for-profit US arts organizations (operating revenue, work space, and number of employees) did not scale similarly with population, and did not scale supra-linearly as other studies have reported for creative endeavors [12]. The linear scaling

of operating revenue with population (exponent β ≈ 1) aligns arts organizations with amenities and individual needs (e.g., recreational opportunities and housing). The analysis argues that the business of art is distinct from the general phenomenon of creativity, at least from the standpoint of the functional scaling of urban systems.

The dissimilar scaling relationships for the three all-arts economic metrics are intriguing. The sub-linear scaling of physical footprint and work force with increasing MSA population in the US is associated with economies of scale typical of infrastructure elements. The narrow range of SAMI values for the operating revenues of museums, dance, theater, and visual arts and design organizations indicates that the economic footprint of these disciplines was well-predicted by population, with less variability in footprint attributable to local factors. Conversely, the wide-ranging SAMI values for music organizations indicate that local contexts play a larger role in determining music's economic footprint in an MSA, a conclusion generally supported by Vanolo [21].

The increase in arts organizations' operating revenue that accompanies larger MSA populations cannot be attributed solely to the greater costs of large metro areas. Cost of living, proxied by real estate and rent costs, increased sub-linearly (β ≈ 0.12) with metro area population, while non-profit arts organization operating revenue increased faster (β ≈ 1). The differences in the population scaling of work space and of real estate value result in an overall reduction in non-profit arts work space in larger, more expensive cities. The sub-linear real estate scaling relationship is beneficial to the arts, as cost of living is of importance to mobile artists [33–35].

More organizations of individual arts disciplines (except dance) were located in large metro areas (>1 M people) than in mid-sized urban systems (50 K–1 M people). However, mid-size metro areas were home to significantly more (ca. 2–7 times) arts opportunities (i.e., on a per capita basis) than were large MSAs, indicating that mid-sized urban systems offer proportionately more arts opportunities than larger systems. However, a more fine-grained examination of dance organizations suggests complexity; child-oriented dance performances (e.g., Tchaikovsky's Nutcracker) are widely popular among the general public and more commonly found in mid-sized cities. New creations, especially by well-known companies, tend to be found in cities with larger populations [36].

Larger US cities generally have greater employment diversity and productivity [37], making them economic drivers of out-sized regional importance. The arts serve as part of this economic engine, while incorporating the socially positive values of inclusion [9] and economic access [26,35]. Urban systems collect creative human capital [18], and pay an "artistic dividend" that includes the export of value (via created arts), the broader downstream effects on ancillary businesses that profit from the traffic generated by arts venues, and the improvement of areas in which artists settle [18,25]. The areas of Philadelphia with a strong arts presence were characterized by greater population growth and a more rapid decline in rates of poverty. This revitalization did not occur via socially disruptive gentrification [38]. Therefore, the presence of arts organizations indicates contributions to much more than simply the economic sustainability of urban systems [6].

Fifteen years ago, artists in the US were concentrated in three dominant creative centers: Los Angeles, New York and San Francisco, with eight "second-tier" metros (Washington DC, Seattle, Boston, Minneapolis-St Paul, Orange County, Miami, Portland, San Diego; [18]). The current study found a similar list of over-performers: Boston, Cleveland, New York, Los Angeles, and Pittsburgh. The 2019 under-performers did not appear on the earlier (2006) list that included St. Louis, Houston, Pittsburgh, Riverside–San Bernardino, San Jose and Tampa [18]. This indicates that the artistic footprint of metro areas may change, though over decadal time scales, a fact supported by non-arts economic trends [15].

A general, conservative strategy for supporting cultural sustainability would focus on the most populous urban areas. Not only are arts opportunities less readily available (i.e., less dense) in large MSAs (>1 M residents), but larger metropolitan areas are also home to disproportionately more college-educated adults. Those individuals with post-high school credentials are much more likely to be consumers of the arts than the

general population [39,40]. In addition, higher income, characteristic of larger urban areas (e.g., [13]), characterizes greater arts event attendance [41,42], although the reader should also consult Poon and Lai [43]. However, a recent study reported the 30% faster growth of the college-educated demographic in small and medium urban areas compared with large (i.e., >1 M inhabitants) areas [44], suggesting that large and medium cities are headed towards achieving parity in arts opportunities.

Urban planners and policy-makers seeking to foster sustainable cultural development should also consider the economies of under-performing MSAs. In this study, when all arts disciplines were considered together, 7 of the 10 most under-performing MSAs were characterized by educational attainment below the national average, and poverty rates exceeding the national average (9 of 10 MSAs); although arts funding may assist in raising the fortunes of the these MSAs, funding for cultural activities will necessarily be part of a larger overall funding strategy.

The arts are economic engines of increased neighborhood prosperity [15] and community identity and cohesion [39]. Cities compete to attract and retain talent in creative occupations. The COVID-19 pandemic has changed the urban landscape, influencing migration to and from New York (and likely other MSAs [40]). In fact, the impact of the viral epidemic may become one of the specific, local factors that influence the degree to which cities obey the general scaling relationship between population and MSA traits. In addition, cities are not likely to all be equally resilient in the face of external shocks [44]; those that have experienced a history of economic contraction (e.g., Detroit, IL, USA) are expected to respond differently than cities with continued economic growth (e.g., Los Angeles, CA, USA). Paradoxically, the COVID-19 pandemic has apparently intensified consumer demand for arts [45], and may elevate productivity in the arts [46,47].

Several other factors merit examination in the future. This study employed economic data from non-profit organizations in the US. Scaling functions for arts funding may vary regionally, with different scaling in different countries. The scaling relationships could also be influenced by the inclusion of for-profit entities in the analysis. This is possibly art-specific, with many for-profit music entities and few for-profit dance entities.

**Funding:** This research was supported via an endowment provided by Lucille Werlinich.

**Institutional Review Board Statement:** Not applicable.

**Informed Consent Statement:** Not applicable.

**Data Availability Statement:** Proprietary data were obtained from SMU DataArts (2019; https://culturaldata.org/). Publicly available datasets were analyzed in this study. These data were obtained from the US Census Bureau (2019; Metropolitan and micropolitan statistical areas. https://www.census.gov/data/tables/time-series/demo/popest/2010s-total-metro-and-micro-statistical-areas.html accessed 30 March 2021); American Community Survey (2020; https://www.census.gov/data/tables/2020/demo/educational-attainment/cps-detailed-tables.html; accessed 16 November 2020); Zillow Home Value Index (2019; https://www.zillow.com/research/data/; accessed 21 March 2021); US Department of Housing and Urban Development (2019; 50th Percentile Rent Estimates https://www.huduser.gov/portal/datasets/50per.html#2019 accessed 21 March 2021).

**Acknowledgments:** Cedric Ceulemans provided valuable comments on an early draft. Daniel Kraemer provided a Python script to automate data summary.

**Conflicts of Interest:** The authors declare no conflict of interest. The funder had no role in the design of the study; in the collection, analyses, or interpretation of data; in the writing of the manuscript, or in the decision to publish the results.

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
