# Peer review of "Cultural Sustainability of US Cities: The Scaling of Non-Profit Arts Footprint with Population"

_sustainability, doi:10.3390/su14074245_

Round 1

Reviewer 1 Report

I do like the main idea of the paper and authors made significant work in this study in order to create the list of indicators of Cultural sustainability.
Then for me it was not clear, why authors decided to concentrate on the period from 2019- I think some logical linking paragraphs must be added and mentioning of this period should be added in the paper.
There is a significant pull of papers that had been used for the development of framework, but it is not clear, how this list of papers have been obtained. Which keywords, databases have been used, inclusion/exclusion cretarie. It is important as based on it authors shows creteria.

Author Response

I do like the main idea of the paper and authors made significant work in this study in order to create the list of indicators of Cultural sustainability.  Then for me it was not clear, why authors decided to concentrate on the period from 2019- I think some logical linking paragraphs must be added and mentioning of this period should be added in the paper. 

The data provided by SMUDataArts is updated yearly, with the dataset analyzed in this paper from 2019 only.  I have added additional text to this effect in the Materials and Methods. These data are pre-pandemic; they present a baseline against which the impact of the external (COVID) shock can be compared. 

There is a significant pull of papers that had been used for the development of framework, but it is not clear, how this list of papers have been obtained. Which keywords, databases have been used, inclusion/exclusion cretarie. It is important as based on it authors shows creteria.

The supporting works presented in the Introduction were chosen to develop the sustainability context for the analysis presented (importance of arts culture to sustainability broadly, and to city arts footprint in particular).  The intent was to outline the context for the study via key papers that build the rationale.

Regarding the literature discovery:  the search began with already-known relevant authors (e.g., Bettencourt, Florida, Markusen), and branched out using scaling, cultural sustainability, urban, arts, culture, arts funding, non-profit, talent, creativity, arts attendance, and cultural cluster as search terms.  The databases most often used were ProQuest Social Science Database, Science Direct, Web of Science.  The seminal papers (e.g., Bettencourt et al., 2007) were also used as a search starting point; papers citing these important works broadened and deepened the search window.

The revised manuscript’s list of supporting literature has been broadened by 25%.

Bettencourt, L. M., Lobo, J., Helbing, D., Kühnert, C., & West, G. B. (2007). Growth, innovation, scaling, and the pace of life in cities. Proceedings of the national academy of sciences, 104(17), 7301-7306.

Reviewer 2 Report

Thank you so much for the opportunity to read the article "Cultural sustainability of cities: the scaling of non-profits arts footprint with population." This paper represents an exciting contribution to a broad discussion to include culture and arts as part of sustainability. In this sense, we attempt to offer some insights to clarify some points:

  1. Philosophical depth and literature review: we suggest including and focusing on the idea about culture as the fourth pillar of sustainability. Although the author explains that he will center on the idea that culture is art, some references on this reduction are necessary, even though to justify the study's limitations.   
  2. Lines 57-60 mention the city's planning uses "big data to improve energy efficiency and quality of life." Its connection with the previous and the following paragraphs is not clear. Also, which are its relationship with the study?
  3. Is there a connection between cultural footprint and the increasing quality of life? Is there a discussion on quality of life as a matter of sustainability? If so, we suggest including it briefly to make the paper's background stronger and use it as part of its conclusions and final discussion (lines 239 and following).
  4. It is not clear to readers that the study is limited to non-profit arts organizations or if it includes for-profit organizations. Explaining it (for example, in lines 96 and ahead) and making clear your decision to use or not use it as a criterion - or even if it is an irrelevant criterion - can help clarify the given conclusions.
  5. An advice that the study is limited to US cities is needed. Show study's limitations are necessary to contribute to future research in different contexts (economically, socially, diverse, for example). For that reason, generalizations made in the introduction and during the discussion should be removed and measured by the scope and limitation of the study (it's not a global study).

Despite these suggestions, again, we consider this paper a piece that has a contribution to the field. It has a solid methodological basis, it's clear, and has the potential to achieve a significant and interdisciplinary audience.

Author Response

Thank you so much for the opportunity to read the article "Cultural sustainability of cities: the scaling of non-profits arts footprint with population." This paper represents an exciting contribution to a broad discussion to include culture and arts as part of sustainability. In this sense, we attempt to offer some insights to clarify some points:  Thank you.  Your detailed comments were helpful.

  1. Philosophical depth and literature review: we suggest including and focusing on the idea about culture as the fourth pillar of sustainability. Although the author explains that he will center on the idea that culture is art, some references on this reduction are necessary, even though to justify the study's limitations.  I have broadened and deepened both the introductory background for context (ca. 50% increase), and the implications in the Discussion.   
  2. Lines 57-60 mention the city's planning uses "big data to improve energy efficiency and quality of life." Its connection with the previous and the following paragraphs is not clear. Also, which are its relationship with the study? Good point.  The text to connect the idea of quality of life with sustainability has been substantially reorganized and expanded.  Energy efficiency was simply one example of the value of big data toward understanding a process (it has been removed).  The connection between arts consumption and self-evaluated happiness is well-known. I cite four studies in the Introduction.     
  3. Is there a connection between cultural footprint and the increasing quality of life? Is there a discussion on quality of life as a matter of sustainability? If so, we suggest including it briefly to make the paper's background stronger and use it as part of its conclusions and final discussion (lines 239 and following).  A robust literature exists that connects arts and sustainability with quality of life.  I’ve added text to the Introduction to address these concerns (e.g., “people in cities that rank higher on sustainability scales reported higher levels of happiness or satisfaction with life”), with four citations.
  4. It is not clear to readers that the study is limited to non-profit arts organizations or if it includes for-profit organizations. Explaining it (for example, in lines 96 and ahead) and making clear your decision to use or not use it as a criterion - or even if it is an irrelevant criterion - can help clarify the given conclusions.  Yes, that is a valuable distinction that needs to be made.  The Materials and Methods now explains that the data source tracks only non-profit organizations.  The Discussion returns to this at the end, considering future lines of research that would examine (i) scaling pre- and post-COVID as a measure of vulnerability; (ii) the degree of spatial homogeneity in scaling (i.e., on a country scale), and; (iii) consideration of the role of for-profit entities, probably varying by arts discipline (for-profit more important for music, for example, much less so for dance).
  5. An advice that the study is limited to US cities is needed. Show study's limitations are necessary to contribute to future research in different contexts (economically, socially, diverse, for example). For that reason, generalizations made in the introduction and during the discussion should be removed and measured by the scope and limitation of the study (it's not a global study).  Good point regarding the data limitation; I have clarified this in the title, and with text at the Discussion’s end acknowledging the existence of lines of research that will provide a better global understanding of arts and city structure as functions of population.  However, I believe that, although the arts data were drawn from US cities, the global generalization regarding the scaling of population and city characteristic is supported by other research. Bettencourt et al. (2007), for example, reported similar scaling relationships for 10 different urban indicators from the US, China, Germany, and the European Union.  Some common, underlying mechanism exists. The MSAs also represent economically, socially, and racially different systems.  Future research can drill down to examine at sub-MSA scales the effects of the differences. 

Reviewer 3 Report

You write from City and metropolitan areas. Please define and elaborate your understanding and definition. 
The basis of your article is to understand cities as a system. But you don't elaborate enough how and why cities can be understood as a system. Just the listing of few reference is not enough. 
What are the elements of the system of cities? What is the role of culture? You use a lot of quantitative data for your research, but the foundation what the role of these figures in the urban system is remains vague. Maybe if you focus on the sub-system of cultural and cultural entrepreneurship it would be more clear. 
You examine revenues and economic figures, but how they are relevant for the inhabitants and how they are connected to for Example Quality of Life or Sustainable Development and SDGs is not really clear from your text. In general you describe interesting findings but they are not contextualized enough to bring a real scientific value.

Author Response

You write from City and metropolitan areas. Please define and elaborate your understanding and definition.   

I thank the reviewer for identifying a source of potential confusion.  The Materials and Methods presents the US Census Bureau’s definition of Metropolitan Statistical Areas (“county-based regions containing at least one urbanized area surrounded by a socially and economically integrated area”). I have also edited the text to refer to MSAs where specificity dictates, since the terms city and metro area, while conceptually overlapping with MSA, have less clear-cut definitions.

MSAs, the unit of analysis in the study, are useful for understanding large scale function (e.g., Bettencourt, 2015), and are relevant to consideration of the economic footprint of the arts.  Other published works have used MSAs as the unit of analysis.  For example, Peach and Petach (2015) used MSAs in the study of the quality of life in cities, Li (2020) used them in connecting the cultural economy and city competitiveness.

Although MSAs are diverse both within and among, this study was based on the premise – for which there exists plenty of support – that MSA characteristics and function varies with population.  These scaling relationships between MSA population and characteristic are similar across cities worldwide (Bettencourt et al. (2007)). 

I have added these and other contextual details to the Introduction (expanded by ca. 50%), and to the implications presented in the Discussion.

The basis of your article is to understand cities as a system. But you don't elaborate enough how and why cities can be understood as a system. 

I agree; MSAs are systems of interacting institutions and processes.  The sub-CCIs (creative and cultural industries) on which the study focused (i.e., the arts disciplines and museums) are themselves interconnected.  

However, this study examined the gross behavior of arts disciplines across a continuum of MSA population, rather than the interactions of arts organizations with other social, political, and economic institutions.  In line with Bettencourt (2013), the present study sought to identify “relations in urban organization and dynamics,” and consider them in the context of sustainability.  I have added clarifying text to the Introduction. 

What are the elements of the system of cities? What is the role of culture? 

The study did not examine the role of culture within urban systems.  Rather, it did essentially the opposite, examining the influence of the urban characteristic (i.e., population) on arts institutions within the system. The research bridges the gap between an easily measured variable (population) and predictions that can guide decision-making and consequent resource allocation.

You use a lot of quantitative data for your research, but the foundation what the role of these figures in the urban system is remains vague. Maybe if you focus on the sub-system of cultural and cultural entrepreneurship it would be more clear.  

Urbanization poses important challenges to understanding growing cities, and to the development of effective sustainability policy (Bettencourt, 2015).  The Introduction has been extensively rewritten to clarify this and other issues.  Culture (customs, social institutions, community history, etc.) is much broader than the focus of this analysis (four arts disciplines plus museums).  Entrepreneurship is also outside the scope of this work. 

You examine revenues and economic figures, but how they are relevant for the inhabitants and how they are connected to for Example Quality of Life or Sustainable Development and SDGs is not really clear from your text. In general you describe interesting findings but they are not contextualized enough to bring a real scientific value.

UN SDG 11 seeks to create environmentally responsible cities with “culturally inspiring environments.”  Quality of life is directly tied to cultural amenities, and arts is an important amenity. The connection between sustainability and arts consumption with self-reported happiness has been solidly made in the Introduction.  If we accept the inclusion of arts in sustainability, a quality-of-life connection also exists. 

In addition some of the relevance for the city inhabitants comes at a step removed, through decisions on the allocation of financial resources.  The study's results should prompt two further lines of policy inquiry: should large MSAs be privileged over mid-size ones, and; should under-performing MSAs receive greater relative funding than over-performing.  I have modified the Discussion accordingly. 

Round 2

Reviewer 1 Report

Dear Author,

Thank you for the revised version of manuscript. The author addressed all the suggestion carefully and enhance the quality of manuscript substantially. I congratulate author for this interesting piece of work.

Reviewer 3 Report

The article has been improved and is now ready for publication